# An In Vitro Antimicrobial, Anticancer and Antioxidant Activity of *N*–[(2–Arylmethylthio)phenylsulfonyl]cinnamamide Derivatives

**DOI:** 10.3390/molecules28073087

**Published:** 2023-03-30

**Authors:** Anita Bułakowska, Jarosław Sławiński, Rafał Hałasa, Anna Hering, Magdalena Gucwa, J. Renata Ochocka, Justyna Stefanowicz-Hajduk

**Affiliations:** 1Department of Organic Chemistry, Medical University of Gdańsk, Al. Gen. J. Hallera 107, 80-416 Gdansk, Poland; jaroslaw@gumed.edu.pl; 2Department of Pharmaceutical Microbiology, Medical University of Gdańsk, Al. Gen. J. Hallera 107, 80-416 Gdansk, Poland; rafal.halasa@gumed.edu.pl; 3Department of Biology and Pharmaceutical Botany, Medical University of Gdańsk, Al. Gen. J. Hallera 107, 80-416 Gdansk, Poland; anna.hering@gumed.edu.pl (A.H.); magdalena.gucwa@gumed.edu.pl (M.G.); jadwiga.ochocka@gumed.edu.pl (J.R.O.)

**Keywords:** cinnamic acid, 2–mercaptobenzenesulfonamide, synthesis, antimicrobial, antibiofilm, MRSA, antitumor activity, antioxidant

## Abstract

Cinnamic acid is a plant metabolite with antimicrobial, anticancer, and antioxidant properties. Its synthetic derivatives are often more effective in vitro than parent compounds due to stronger biological activities. In our study, we synthesized ten new *N*–(4–chloro–2–mercapto–5–methylphenylsulfonyl)cinnamamide derivatives, containing two pharmacophore groups: cinnamic acid moiety and benzenesulfonamide. The antimicrobial activity of the obtained compounds was estimated using different types of Gram-positive and Gram-negative bacteria, fungus species of *Candida albicans*, as well as clinical strains. The compounds were evaluated on biofilm formation and biofilm formed by *Staphylococcus* clinical strains (methicillin–resistance *S. aureus* MRSA and methicillin–resistance coagulase–negative *Staphylococcus* MRCNS). Furthermore, blood bacteriostatic activity test was performed using *S. aureus* and *S. epidermidis*. In cytotoxic study, we performed in vitro hemolysis assay on domestic sheep peripheral blood and MTT [3–(4,5–dimethylthiazol–2–yl)–2,5–diphenyltetrazolium bromide] assay on human cervical HeLa, ovarian SKOV-3, and breast MCF-7 cancer cell lines. We also estimated antioxidant activity of ten compounds with 2,2-diphenyl-1-picrylhydrazyl (DPPH) and 2,2′–azino–*bis*(3–ethylbenzthiazoline–6–sulfonic acid) (ABTS) assays. Our results showed a significant antimicrobial activity of the compounds. All of them were active on *Staphylococcus* and *Enterococcus* species (MIC was 1–4 µg/mL). The compounds **16d** and **16e** were the most active on staphylococci clinical strains and efficiently inhibited the biofilm formation and biofilm already formed by the clinical staphylococci. Moreover, the hemolytic properties of the tested compounds occurred in higher quantities (>32.5 µg/mL) than the concentrations that inhibited both the growth of bacteria in the blood and the formation and growth of biofilm. The results of MTT assay showed that compounds **16c**, **16d**, **17a,** and **17d** demonstrated the best activity on the cancer cells (the IC_50_ values were below 10 µg/mL). Compound **16f** was the least active on the cancer cells (IC_50_ was > 60 µg/mL). Antiradical tests revealed that compounds **16f** and **17d** had the strongest antioxidant properties within the tested group (IC_50_ was 310.50 ± 0.73 and 574.41 ± 1.34 µg/mL in DPPH, respectively, and 597.53 ± 1.3 and 419.18 ± 2.72 µg/mL in ABTS assay, respectively). Our study showed that the obtained cinnamamide derivatives can be used as potential antimicrobial therapeutic agents.

## 1. Introduction

The search for medicine that would act on many diseases simultaneously began thousands of years ago and is still continuing [1]. Currently, the most commonly used method is so-called polytherapy, where several medications are administered at the same time. There have been attempts to integrate molecules of several drugs into one molecule, and hybrid compounds have become more and more interesting for scientists due to their many beneficial properties, especially when these hybrids are composed of naturally occurring compounds. The combination of a plant metabolite—cinnamic acid with sulfonamides—used in therapy is one of the examples. The cinnamic acid is a part of many important biochemical pathways and is commonly found in the plant world. It can be found in *Cinnamomum cassia* (Chinese cinnamon) or *Panax ginseng* [2,3] and it has, inter alia, antimicrobial [4,5], anticancer [6,7], and antioxidant [8,9] properties. Its structure includes a benzene ring, an alkene double bond, and an acrylic acid functional group. The biological and pharmacological properties of cinnamic acid itself and its derivatives may result from the presence of α,β-unsaturated bond, which is responsible for their potential therapeutic effect in cancer treatment [10,11]. Cinnamic acid has a cytoprotective effect on nerves in neurogenerative diseases [12] and an anti-inflammatory effect (at 200 mg/kg/day) [13]. In addition, it significantly reduces the body weight of obese rats and acts as an antihypertensive agent inhibiting angiotensin serum converting enzyme (ACE) activity (at 30 mg/kg/day) [14]. It also shows a cardioprotective profile by preventing vasoconstriction and reducing the risk of complications of hypertension [14]. Cinnamic acid displays antidiabetic activity by improving glucose tolerance in vivo (at 10 mg/kg/day) and stimulating insulin secretion in vitro [15,16].

Furthermore, cinnamic acid is also known as an antimicrobial agent, and its combination with sulfonamides may result in an increased effect of the obtained hybrids on bacteria. Sulfonamides have been popular for many decades and used in medicine [17] as antibacterial (Sulfacetamide), analgesic (Celecoxib), anti-inflammatory (Sulfasalazine), diuretic (Furosemide), or antiepileptic drugs (Zonisamide). The first mentions of the cytotoxic activity of sulfonamides appeared in the 1980s, and the interest in their potential use in cancer treatment increased with the discovery of carcinogenesis processes [18]. Profiles of the biological activity of cinnamic acid and sulfonamide compounds resulted in an increased interest in hybrid molecules based on their structures [19,20,21]. Cinnamic acid derivatives have been evaluated in a few clinical trials [22]. Belinostat–(*E*)–*N*–hydroxy–3–[4–(*N*–phenylsulfamoyl)phenyl]acrylamide (PXD101) is a novel HDAC inhibitor (IC_50_ is 27 nM). It shows activity in cisplatin-resistant tumors and induces autophagy [23]. Panobinostat–(*E*)–*N*-hydroxy–3–[4–({[2–(2–methyl–1*H*–indol–3–yl)ethyl]amino}methyl)phenyl]acrylamide (LBH589) is another example of a new broad-spectrum HDAC inhibitor (IC_50_ is 5 nM). It induces autophagy and apoptosis and effectively interferes with HIV latency in vivo (phase 3) [24].

It needs to be highlighted that in the development of novel antibacterial drugs, the main focus is placed on the multiple resistance situations, which can occur through different mechanisms: mutations resulting in multidrug resistance, efflux of a drug by pumps, loss of porin protein and reduced uptake of an antibiotic drug, and other mechanisms [25,26,27,28]. Therefore, it makes sense to design compounds that contain different pharmacophores, such as hybrid drugs, to circumvent different resistance pathways or cross-resistance.

This study is a continuation of our research on hybrid compounds [29,30,31,32]. Following the promising results of research on derivatives containing the 2-mercaptobenzenesulfonamide group in the scaffold and literature reports, new *N*–(4–chloro–2-mercapto–5–methylphenylsulfonyl)cinnamamide derivatives, containing two pharmacophore groups—cinnamic acid moiety and benzenesulfonamide—were designed and synthesized in this work (Figure 1). The aim of our study was to evaluate the microbiological, anticancer, and antioxidant activities of the obtained hybrid compounds.

## 2. Results and Discussions

### 2.1. Chemistry

The synthesis of new cinnamamide derivatives was carried out in four stages. In the first stage, 4–chloro–2-mercapto–5-methylbenzenesulfonamide **1** was synthesized in the reaction of alkaline hydrolysis potassium of 6–chloro–7–methyl–1,1–dioxo–1,4,2–benzodithiazine–3-thiolane in the presence of 99% hydrazine hydrate in 90% yield [33]. In the second step, (**1**) was substituted with 1–methylnaphthalene or 6–chloropiperonyl, yielding 4–chloro–5–methyl–2–[(naphthalen–1–yl)methylthio]benzenesulfonamide **2** and 4–chloro–2–[(6–chlorobenzo[*d*][1,3]dioxol–5–yl)methylthio]–5-methylbenzenesulfonamide **3** as starting materials for further reactions. In the third step, the benzaldehyde derivatives (**4**–**9**) were reacted with the appropriate malonic acid to obtain cinnamic acid derivatives (**10**–**15**) [34,35,36]. The last step was the synthesis of two series of target compounds. The first series was *N*–{4–chloro–5–methyl–2–[(naphthalen–1–yl)methylthio]phenylsulfonyl}cinnamamide derivatives (**16a**–**16f**) [37]. They were obtained as a result of the reaction of compound **2** with appropriate cinnamic acid derivatives (**10**–**15**) in the methylene chloride solution at room temperature for 24 h, with the presence of 1–ethyl–3–(3–dimethylaminopropyl)carbodiimide (EDCI) condensing reagent and 4–dimethylaminopyridine (DMAP) in 39–59% yield. The synthesis of the second series of derivatives (**17a**–**17d**) proceeded in the same way as for substrate (**3),** which was reacted with the appropriate cinnamic acid derivatives (**10**–**15**) at room temperature for 24 h in the methylene chloride solution, with the presence of EDCI and DMAP. The reaction yield was in the range of 39–64% (Figure 1).

The structures of the final compounds described in this paper were confirmed by spectroscopic data and elementary analyses (C, H, N). While analyzing the IR spectra of the obtained products (**16a–16f**, **17a–17d**), the occurrence of absorption bands originating from the vibrations stretching the NH groups in the range of 3152–3368 cm^−1^ was noticed. The range of 2858–2981 cm^−1^ was characterized by the presence of absorption bands coming from the stretching vibrations of the CH (aliph.) group, while the bands of the C=O groups were in the range of 1665–1719 cm^−1^. Within the range of 1436–1633 cm^−1^, bands of NH deformation vibrations and C=C vibrations were observed. In the dactyloscopic range, there were also bands corresponding to the asymmetric vibrations of the SO_2_ group at approx. 1350 cm^−1^ and the symmetric vibrations at 1170 cm^−1^. A common feature of the ^1^H NMR spectra for the synthesized products was the presence of a sharp singlet at ca. 2.5 ppm, resulting from the protons of the methyl group in the 5-position of the benzene ring. Attention was also drawn to the characteristic doublets with coupling constants of 15.6–16.1 Hz (*trans* isomers), originating from alkenyl protons, appearing at about 6.5 ppm and 7.5 ppm. All spectra also contained two singlets resulting from protons located in positions 3 and 6 of the benzene ring, at values of approximately 7.75 ppm and 8.0 ppm, respectively. On the other hand, a weak signal in the form of a singlet, appearing at about 12.5 ppm, came from the proton of the NHSO_2_ group. The difference in ^1^H NMR spectra between derivatives containing a naphthylmethylthiol substituent and derivatives containing a 6–chloropiperonylthio group is the presence of singlets derived from protons of methylene groups. Compounds with a naphthyl substituent contain only one singlet, which gives the S–CH_2_ group approx. 4.4 ppm, while in the spectra for compounds with 6–chloropiperonyl on the sulfur atom, an additional signal is visible, which comes from the protons of the O–CH_2_–O group from the dioxolane ring at ca. 6 ppm.

### 2.2. Biological Studies

#### 2.2.1. Antimicrobial Activity

##### Minimum Inhibitory Concentration Determination

Our previous study on 4–chloro–2-mercapto–5–methylbenzenesulfonamides derivatives showed that these compounds have antimicrobial activity [30]. In this study, we evaluated the effect of cinnamamide derivatives on reference strains of Gram-positive bacteria—*Staphylococcus aureus* ATCC 6538, *Staphylococcus epidermidis* ATCC 14990, *Enterococcus hirae* ATCC 10541, *Enterococcus faecalis* ATCC 51299, *Bacillus subtilis* ATCC 6633, *Corynebacterium diphtheriae*—and Gram-negative bacterial strain of *Escherichia coli* ATCC 8739. *Candida albicans* ATCC 10231 was also involved in this study (a fungal species). The results of the antimicrobial activity of the tested compounds are presented in Table 1.

The compound crystals were dissolved in DMSO and diluted in sterile water (DMSO concentration did not influence the viability of bacterial strains). They showed high inhibitory activity (MIC) against reference Gram-positive cocci in the range of 1–8 µg/mL and low activity against other reference Gram-positive microorganisms (*Bacillus subtilis*, *Corynebacterium diphtheriae*) and *Escherichia coli* (concentrations equal to or higher than 62.5 µg/mL). Co-trimoxazole (for bacteria), Ketoconazole (for fungi) were used as reference compounds. The derivatives most active on *Staphylococcus* sp. and *Enterococcus* sp. were **16b**, **16c**, **16d**, **17a**, and **17c**. Their MIC values were in the range of 1–2 µg/mL. Furthermore, the results for compound **16a** were consistent with the data obtained in our previous study [30].

In the next step, we evaluated the compounds on clinical methicillin-resistant *S. aureus* (MRSA) strains (Figure 2) and methicillin-resistant coagulase-negative *Staphylococcus* (MRCNS) (Figure 3). The group of CNS includes nonpathogenic or opportunistic pathogens, while coagulase-positive staphylococci are represented by *S. aureus*. About 30% of the human population carries *S. aureus* in the upper respiratory tract. Infections caused by *S. aureus* include all purulent skin conditions (boils, abscesses, folliculitis) and nosocomial infections. The strains isolated from hospitals are usually antibiotic-resistant, where MRSA strains are the majority. *Staphylococcus epidermidis* (like other coagulase-negative *Staphylococcus*) is included in the microflora of human skin and can easily cause surface contamination after surgical procedures.

In our study, we used clinical *Enterococcus* species (Figure 3), which are commensal to the human and animal digestive tract. *Enterococcus* sp. (*E. faecalis* and *E. faecium*) are the main pathogens responsible for urinary tract infections, bacteremia, intra-abdominal infections, and endocarditis. They are responsible for nosocomial infections, which, in combination with the rapidly growing resistance to aminoglycosides, glycopeptides, and other antibiotics, create therapeutic difficulties. The results of the antimicrobial activity of all the tested compounds on clinical strains, including MRSA, MRCNS, and *Enterococcus* sp., are presented in Figure 2 and Figure 3, and Appendix A.

Values effective against *S. aureus* MRSA and MRCNS were generally in the range of 1–4 µg/mL, with the exception of compound **17a**, which showed activity in the range of 8–16 µg/mL against the following strains: MRSA12673, MRSA21804, MRSA18532, MRSA12647, and MRSAN315 VISA. Compound **17b** showed MIC > 125 µg/mL for MRSA12673, MRSA21804, MRSA12647, MRSA18532, and MRSAN315 VISA. The tested compounds showed activity against MRCNS strains in the range of 1–4 µg/mL. On the other hand, the activity of the tested compounds against clinical strains of *Enterococcus* was more diverse than against staphylococci and was in the range of 1–32 µg/mL. The activity of compounds **17b** and **17d** against *Enterococcus* sp. 3937152 was outside of the tested concentration range (MIC > 125 µg/mL). A similar situation was for compound **17b** against *Enteroccoccus* sp. 3934825 and *Enterococcus* sp. 12848. Minimum bactericidal concentrations (MBC) for most of the tested compounds were equal to or above 125 µg/mL (data not shown).

Inhibitory concentration values obtained in our study for compound **16a** and clinical strains, including MRSA and MRCNS, were similar to those obtained by Żołnowska et al. [30]. Chemical modifications of compound **16a** allowed obtaining derivatives with high activity against the tested clinical strains. The most active compounds were **16b**, **16d**, **16e,** and **16f** with 1–naphthalen at the R^2^ position. However, in the experiments with clinical strains of *Enterococcus* sp., the compounds were characterized by different activities, where **16a** and **16c** were the most active. This may be related to different mechanisms of antibiotic resistance present among these isolates. It should be noted that compounds substituted in R^2^ with 1–naphthalen showed greater antibacterial activity against the tested strains than the derivatives with 6–chloropiperonyl. Furthermore, the derivatives with 1–naphthalen have the same substituents (bromine, chlorine, fluorine, and nitro group) in R^1^ position as those with 6–chloropiperonyl, which indicates that the type of substituent in R^2^ determines the strength of the antimicrobial activity of compounds derived from cinnamic acid and sulfonamide. Simultaneously, the presence of chlorine in cinnamamide hybrids can also be responsible for this effect, which is shown in the results obtained for compounds **17a**–**d**, where compound **17c** showed higher activity than the other **17** derivatives.

Malheiro et al. studied the activity of fifteen selected cinnamic acid derivatives on *S. aureus* NCTC 10788, *E. coli* NCTC 10418, and *E. hirae* NCTC 13383. All the bacteria were inhibited by cinnamaldehyde in a concentration range of 3–8 mM. The concentrations of 8–10 mM killed *S. aureus*. Compounds such as cinnamic, hydrocinnamic, α–methylhydrocinnamic, α–methylcinnamic, and α–fluorocinnamic acids inhibited bacterial growth at concentrations of 15–25 mM. The inhibitory effect of cinnamic acid derivatives on the growth of bacteria was confirmed by the demonstration of an increased doubling time or increased duration of the lag phase at a concentration of 5 mM, despite high MIC values (>25 mM) for most of the tested compounds [5].

Huang et al. also reported that methyl *trans*-cinnamate increased lag phase length and decreased exponential phase at 3.1 mM for *E. coli* and completely inhibited growth at 3.1 mM for *S. aureus* and 6.2 mM for *E. coli* [38].

All the reported results for esters of cinnamic acid and isobutyl cinnamate showed their potent antibacterial activity. The MIC values of the compounds ranged between 43 and 301 μM against Gram-positive (including *S. aureus*) and Gram-negative (including *E. coli*) bacteria [4,39].

Interestingly, sinapyl amides also showed selective growth inhibition on Gram-positive cocci bacteria (MIC between 171 and 696 μM), while being much less effective inhibitors of *C. albicans* (MIC between 1.36 and 1.53 mM) [4,40].

Synthetic cinnamic hybrids containing cephem, cinnamic aldehyde, and 2–hydroxy-3–propylamine were the most active against *S. epidermidis* A24548 (MIC = 22 nM) and *S. haemolyticus* A21638 (MIC = 87 nM) [41].

##### Inhibition of Biofilm Formation by Cinnamic Acid Derivatives

Biofilm is a matrix-covered cell-accumulating structure that is highly resistant to chemical permeation, which can lead to difficult-to-treat biofilm-borne infections. The study on the effect of the tested chemical compounds on the biofilm produced by selected clinical strains of staphylococci was introduced in our work in two systems: the impact on biofilm formation and on the biofilm already formed. The effect of the compounds on biofilm was tested at two concentrations: double (2 × MIC) and quadruple (4 × MIC) minimum inhibitory concentration. In this step of our study, we selected the derivatives most active on MRSA bacteria. We chose the compounds with 1–naphthyl substituent in R^2^—**16a**, **16b**, **16d**, **16e**, and **16f**. We also tested **17c**, which has 6–chloropiperonyl and chlorine atom in R^2^ and R^1^, respectively. Compound **17c** is a derivative with lower inhibitory activity (lower MIC) on MRSA strains; thus, it can be in opposition to compound **16d**, which has 1–naphthyl in R^2^ and chlorine in R^1^. The results of the effect of the selected compounds on biofilm formation are presented in Figure 4 and Figure 5. The effect of the compounds on the biofilm already formed is shown in Figure 6 and Figure 7.

Figure 4 shows the effect of the tested compounds on biofilm formation by *S. aureus* clinical strains. The effectiveness of the derivatives on inhibiting the biofilm-forming capacity of MRSA strains varied widely from 61% (compound **17c** at 2 × MIC against MRSA21804) to 98% (compound **17c** at 2 × MIC against MRSA13318) compared with the control. The compound **16a** (at 4 × MIC) showed the weakest inhibitory properties. Comparing the obtained results, compound **16d** is more active than compound **17c** against the biofilm formed by the tested strains.

Figure 5 shows the effect of the tested compounds at 2 × MIC and 4 × MIC concentrations on biofilm formation by clinical strains of coagulase-negative *Staphylococcus*. Most of the compounds at both concentrations were more than 90% effective in preventing biofilm formation of the MRCNS strains compared with the control. Compounds **16d** and **16e** at 2 × MIC and 4 × MIC concentrations showed biofilm inhibition efficacy of 82–84% against MRCNS1600 and MRCNS16248 compared with the control.

The chemical compounds showed differential activity further limiting biofilm formation by MRSA strains (Figure 6). The degree of inhibition ranged from 40% to 86% in relation to the control. The most sensitive strain turned out to be MRSA21804 (inhibition from 67% to 86%), and the most resistant was MRSA12677 (inhibition of about 40% for **17c** at 2 × MIC and of about 53% for **16e** at 2 × MIC and 4 × MIC) in relation to the control.

Figure 7 shows the effect of the selected compounds on 24 h biofilm formed by MRCNS strains. The tested concentrations of the compounds (2 × MIC and 4 × MIC) showed a high degree of inhibition for biofilm development. Most results revealed inhibition in the range of 70–97% for all the compounds. Only the MRCNS1600 strain turned out to be more resistant to compounds **16a**, **16e,** and **17c** (inhibition from 28% to 68%) compared with the control. Compound **16a** turned out to be less active than the tested derivatives (**16d**, **16e**, **17c**). The obtained results showed that compounds **16d**, **16e**, and **17c** have a similar level of activity on the biofilm formed by the tested MRCNS strains.

Żołnowska et al. showed that selected compounds at concentrations from 0.5 × MIC to 4 × MIC inhibited biofilm formation (35% to 91%) by selected strains of MSSA and MRSA [30]. In our studies, compound **16a** and its new derivatives significantly inhibited biofilm formation at concentrations of 2 × MIC and 4 × MIC for MRSA (61–98% inhibition for most strains). Additionally, we showed high antibiofilm activity of the derivatives on MRCNS strains (over 90% inhibition of biofilm formation). Our data show that the compounds have a high inhibitory activity on the discoloration of the biofilm already formed by the MRSA and MRCNS strains as well as on the formation of biofilm by staphylococci strains. It can be assumed that compounds **16d** and **16e** can be practically used as potential antibacterial preparations and coatings in medical devices or medical implants, where staphylococcal biofilm is very often difficult to eliminate [42]. The compounds **16d** and **16e** consist of chlorine and fluorine with methoxy group in R^1^, respectively, and 1–naphthalen in R^2^ position. In the case of compound **17c**, chlorine is also present in R^1^ position, but in R^2^, it is 6–chloropiperonyl; however, this compound is only slightly less active than compound **16d**. This confirms that the presence of chlorine in R^1^ position can play an important role in the antibacterial activity of cinnamamide derivatives.

Malheiro et al. studied the effect of cinnamic acid derivatives on the quorum sensing system of *C. violaceum*, homologs of the LuxI/LuxR system. Quorum sensing is a way of communication between microorganisms, which is important in the formation of biofilm. The study showed that cinnamic acid, cinnamamide, α–methylcinnamic acid, 4–chlorocinnamic acid, 4–nitrocinnamic acid, and 3,4–(methylenedioxy)cinnamic acid were able to completely inhibit quorum sensing at 5 mM as well as cinnamyl alcohol, allyl cinnamate, and methyl *trans*-cinnamate at 1 mM [5].

##### Blood Bacteriostatic Activity Tests

The manifestation of the bactericidal properties of chemical compounds is low minimum bactericidal concentration (MBC) values, which allows many groups of researchers to check the killing rate over time (time killing tests). It is an important element in determining the strength and speed of action of a compound on bacteria. Such studies are usually carried out in bacterial media, without additional factors influencing the results. Since our compounds show bacteriostatic power, which has been demonstrated in study on microdilutions in medium, we decided to check whether their inhibitory properties are preserved in sheep blood under in vitro conditions.

This step was carried out on two types of bacteria, *S. aureus* ATCC 6538 (Figure 8 and Table 2) and *S. epidermidis* ATCC 14 990 (Figure 9 and Table 3).

The number of bacteria at the initial time (t_0_) was 1.30 × 10^8^ CFU/mL for *S. aureus* and 1.70 × 10^7^ CFU/mL for *S. epidermidis*. Live bacterial cells were assayed after 24 h of incubation at 37 °C. The presented results showed that in the presence of blood, the compounds concentrations of 1, 2, 4, and 8 µg/mL exerted bacteriostatic properties against two standard strains of *Staphylococcus*. Bacterial controls without compounds grew unhindered and after 24 h (t_24_) were 2.28 × 10^9^ for *S. aureus* and 5.10 × 10^8^ CFU/mL for *S. epidermidis*. It should be noted that the compounds within the tested range of concentrations exhibited inhibitory properties despite the presence of erythrocytes, leukocytes (e.g., macrophages, neutrophils), and whole plasma in the blood samples. This means that the blood components did not affect the inhibitory properties of the compounds in the tested range. It can be assumed that the tested compounds were not related to any fraction of the blood. Attention should be paid to the *S. epidermidis* tests and the difference between the number of bacteria at t_0_ and after 24 h of incubation for each compound. The number of bacteria at t_0_ was 1.70 × 10^7^ CFU/mL and after 24 h it decreased to, e.g., 9.40 × 10^6^ CFU/mL for compounds **16e** and 7.80 × 10^6^ CFU/mL for compound **17c**. It can be assumed that the tested compounds after being added to blood samples prevented bacteria from multiplying. However, under these conditions, the phagocytic cells take up the bacteria. The number of phagocytes is limited, hence a similar number of bacteria was visible after 24 h in all the tested samples with the compounds. On the other hand, in tests with *S. aureus*, the number of bacteria between t_0_ and t_24_ did not show much difference. This can be explained by much greater variety of mechanisms of action of *S. aureus* against phagocytic cells than of *S. epidermidis*. This can be integrated with the bacterial cell compounds, e.g., protein A, as well as enzymes that bacteria can secrete outside the cell, e.g., coagulase and staphylokinase. Additional studies should be conducted to check how the tested compounds affect the enzymes secreted by *S. aureus*.

#### 2.2.2. Cytotoxic Activity

##### Hemolysis Assay

The blood cell hemolysis test is performed by disrupting the cell membrane [43]. Most researchers study the hemolytic (cytotoxic) effects of chemical compounds on isolated erythrocytes [44]. Such study does not allow to fully assess the activity of compounds and the influence of other blood components on the effect of erythrocyte lysis. To perform bacteriostatic stability tests in our work, we first conducted a hemolysis assay for all the tested compounds. The results from this assay are shown in Figure 10.

Figure 10 presents hemolytic properties of all the tested compounds on the blood of domestic sheep in vitro after 24 h of incubation at 37 °C. The tested compounds at concentrations of 0.5–16 μg/mL did not show hemolytic properties. We observed hemolysis at the level of 12–20% for compounds **16a**, **16b**, **16e**, **16f**, **17c,** and **17d** at a concentration of 31.25 μg/mL. All the compounds showed hemolysis at concentrations of 62.5 and 125 μg/mL. Their activity levels ranged from 9% for compound **17b** at 62.5 μg/mL to 43% for compound **16a** at 125 μg/mL. The positive control was 1% triton-x 100. In relation to blood bacteriostatic activity results, we concluded that the hemolytic properties of the tested compounds occurred at concentrations higher than the concentrations that inhibited both the growth of bacteria in the blood and the formation and growth of biofilm. The tested compounds in the concentration range of 0.5–8 µg/mL most likely did not bind to blood components and maintained their bacteriostatic properties (acted selectively on bacterial cells). The compound concentrations equal to or higher than 32 µg/mL caused interaction with erythrocytes in peripheral blood, leading to hemolysis. It can be assumed that at higher concentrations, the tested compounds destabilized the cellular membrane of blood cells, causing the formation of holes in the membrane and cracking of the cells.

##### Anticancer Activity

To estimate the cytotoxic activity of all the tested compounds on cancer cells we used an MTT assay in our study. The IC_50_ values of the compounds on HeLa cells were in the range of 8.49–62.84 µg/mL. The most active compounds were **16d**, **17a**, and **17d** (IC_50_ < 10 µg/mL). In the experiments with SKOV-3 cells, we obtained IC_50_ values in the range of 7.87–70.53 µg/mL, and in this case, the most active compounds were **16c** and **16d**. Furthermore, the results obtained on MCF-7 cells showed IC_50_ values in the range of 11.20–93.46 µg/mL. The compounds **16c**, **16d**, **17a**, and **17c** had the highest activity within the tested group. On the other hand, compound **16f** with nitro and 1–naphthyl substituents showed the weakest activity on all the tested cell lines (Table 4).

We also observed the cytotoxic effect of the compounds on noncarcinogenic cell line—HEK-293. The results were similar to those obtained for the cancer cells. The IC_50_ values were in the range of 6.49–94.46 µg/mL.

In the next step, we calculated the selectivity index (SI) for all the compounds (Table 5) according to the formula:SI = IC_50non-cancer cells_/IC_50cancer cells_
SI ≥ 10 is required for a compound to be considered selective for cancer cells.

The selectivity index (SI) results showed that there was no selectivity effect of all the compounds on cancer cell lines. The values of SI were between 0.5 and 1.5.

Antitumor activities of various cinnamic acids and sulfonamide hybrids were examined in a few studies. Luo et al. synthesized a series of novel cinnamamide derivatives and evaluated their biological effect on mouse melanoma B16–F10 cell line in vitro. Among the tested compounds, two of them—(*E*)–3–(4–fluorophenyl)–*N*–(4-fluorophenylsulfonyl)acrylamide and (*E*)–3–(4–bromophenyl)–*N*–(4–fluorophenylsulfonyl)acrylamide—had a significant anticancer effect with IC_50_ values of 0.8 and 1.2 µg/mL, respectively [37]. Next study revealed that among other cinnamamide hybrids, compound (*E*)–3–(benzo[*d*][1,3]dioxol–5–yl)–*N*–(phenylsulfonyl)acrylamide displayed high antiproliferative activity on human breast cancer MCF-7 cells with IC_50_ value of 0.17 µg/mL [20]. In other experiments with *N*–(2–arylmethylthio–4–chloro–5–methylbenzenesulfonyl)amides, molecular hybrids were obtained by combining the 2–mercaptobenzenosulfonamide fragment with a cinnamic acid residue [29,30]. The biological results of the study showed that the compounds tested on human cervical HeLa, breast MCF-7, and colon HCT-116 cell lines did not exert a significant cytotoxic effect (IC_50_ values were in the range of 42–170 µM) [29,30].

In this study, the cytotoxic activities of the cinnamamide derivatives were very moderate. The best activity on the cancer cells was observed for compounds **16c** and **16d**—with fluorine and chlorine substituents in R^1^, respectively, as well as with 1–naphthyl substituent in R^2^. Compounds **17a** and **17d,** with bromine and nitro group in R^1^, respectively, and 6–chloropiperonyl in R^2^ had also a stronger cytotoxic effect on cancer cells. In all these cases, the IC_50_ values were below 10 µg/mL (≤20 µM). The presence of this kind of substituents in the compounds can explain their activity on cancer cells.

#### 2.2.3. Antiradical Activity

The antiradical activity of the tested cinnamic acid derivatives was analyzed with the DPPH and ABTS tests. The results expressed as IC_50_ are shown in Figure 11 and Table 6.

The research indicated weak antiradical properties for all the analyzed cinnamic acid derivatives compared with the well-known chemical compound of ascorbic acid (IC_50_ 10.78 ± 0.11 and 18.99 ± 0.03 µg/mL for DPPH and ABTS analysis, respectively). The most promising antiradical properties for DPPH radical scavenging ability were demonstrated by compounds **16f** and **16d** (IC_50_ 310.50 ± 0.73 and 348.21 ± 0.29 µg/mL, respectively), while in the ABTS test, IC_50_ values of **17d**, **17c**, **17a**, and **16f** were lower than 600 µg/mL (IC_50_ 419.18 ± 2.72; 496.63 ± 0.48; 569.99 ± 7.9 and 597.53 ± 1.3 µg/mL, respectively). The statistical analysis indicated that the ABTS test showed no statistically significant differences between the IC_50_ of cinnamic acid derivatives, while for all the compounds and ascorbic acid, statistical differences were observed. For the DPPH test statistical differences were observed for all the analyzed cinnamic hybrids and ascorbic acid (*p* > 0.05).

Weak antiradical properties of the analyzed cinnamic acid derivatives indicate that they have a minor ability to transfer unbound electrons and protect against free radicals. It should be emphasized that the above results were obtained in vitro, whereas in vivo the antioxidant capacity of the analyzed compounds may be much higher. Derivatives of cinnamic acid with nitrogen oxide moiety (**16f** and **17d**) produced promising results. Their antioxidant properties are probably connected with the acylvinyl group in the hybrid. For compounds **17a**–**17d,** an additional piperonyl reactive group can be responsible for better results in the ABTS test [45]. Compounds **16d** and **17d** exhibited both stronger anticancer and small antiradical properties, although further research should be conducted if the anticancer activity is connected to the radical scavenging ability of the analyzed compounds. Compound **16f** exhibited promising antioxidant results, while there was only a minor activity seen against cancer lines (Table 6 and Figure 11). This observation suggests that this compound has properties for screening cells for free radicals, which warrants further analysis in subsequent studies. The analyzed cinnamic acid derivatives probably exhibit other than the antiradical mechanism of action on bacteria and cancer cells (despite probably compounds **16d** and **17d**), which should be determined in further research.

## 3. Materials and Methods

### 3.1. Antimicrobial Study

#### 3.1.1. Materials

Sterile sheep blood defibrinated (GrasoBiotech). Co-trimoxazole (composition sulfamethoxazole 400 mg/trimethoprim 80 mg; WZF Polfa Poland). Ketoconazole (WZF Polfa Poland). Bacteria: *Enterococcus hirae* ATCC 10541 and *Enterococcus faecalis* ATCC 51299, *Staphylococcus epidermidis* ATCC 14990, *Staphylococcus aureus* ATCC 6538, clinical strains *Staphylococcus aureus* MRSA, clinical strains *Staphylococcus* sp. (MRCNS), clinical strain *Staphylococcus epidermidis* MRSE, clinical strain *Enterococcus* sp., *Corynebacterium diphtheriae* ZMF, *Bacillus subtilis* ATCC 6633, *Escherichia coli* ATCC 8739, *Candida albicans* ATCC 10231. Brain-heart infusion broth (BHI, Becton Dickinson) for *Enterococcus hirae* ATCC 10541 and *Enterococcus faecalis* ATCC 51299, BHI supplement with 10% bovine serum for *Bacillus subtilis*, *Corynebacterium diphtheriae* ZMF, *Staphylococcus epidermidis* ATCC 14990, *Staphylococcus aureus* ATCC 6538, clinical strains *Staphylococcus aureus* MRSA, clinical strains *Staphylococcus* sp. (CNS), clinical strain *Staphylococcus epidermidis* MRSE grew in Mueller-Hinton broth (MH cation-adjusted, Becton Dickinson) in an aerobic atmosphere at 37 °C for 48 h, in an aerobic atmosphere at 37 °C for 72 h. *Candida albicans* grew in Sabourauda broth (Sb, Becton Dickinson), in an aerobic atmosphere at 37 °C for 72 h. After determination of the bacterial viability, BHI blood agar plates, MH agar plates, and Sb agar plates were used.

#### 3.1.2. Minimum Inhibitory Concentration Determination

The MIC determination for the reference and clinical strains was performed based on the methodology described by Kula et al. [46,47]. The dry test samples were dissolved in dimethyl sulfoxide (DMSO) and diluted in water, resulting in a finale concentration of about 500 µg/mL. These solutions were diluted and added to the first well of each microtiter line. Dilution in geometric progression was done by transferring the mixture/dilution (100 μL) from the first to the twelfth well. An aliquot (100 μL) was discarded from the twelfth well. The final concentration of the samples used in the antimicrobial activity assay ranged from 125 to 0.006 µg/mL. Co-trimoxazole (for bacteria), with the composition sulfamethoxazole 400 mg/trimethoprim 80 mg was used as a reference compound in the concentration range of 20 000/4000 to 1/0.2 µg/mL. Ketoconazole (for fungi) was used as a reference compound in the concentration range of 125 to 0.006 µg/mL. Tests were incubated in adequate conditions described by Kula et al. [47,48]. Diluent concentration had no effect on the activity of the tested compounds. All experiments were carried out three times.

#### 3.1.3. Inhibition of Biofilm Formation by the Cinnamic Acid Derivatives

Determination of the biofilm (prior to biofilm and post-biofilm formation) inhibitory concentration were performed through adaptation of the previous work [48] with a modification. The overnight bacteria culture diluted in TSB (tryptic soy broth) supplemented with 2% glucose to 108 CFU/mL and compound concentrations of 2 × MIC and 4 × MIC was dispensed into each well of a 96-well plate. The final volume of the mixture was 200 µL. In the post-biofilm formation, biofilm was formed for 24 h at 37 °C, non–adherent cells were removed, and compounds at the concentrations of 2 × MIC and 4 × MIC were added into each well. After 24 h incubation at 37 °C, the contents of each well were discarded and washed 3 times with sterile deionized water in order to remove non–adherent cells. The biofilm was fixed with 2% formaldehyde (0.5 h), and then stained with 0.1% (*v*/*v*) crystal violet solution for 1 h at 37 °C; then the plates were rinsed with water until clean drops were obtained. The stained biofilm was dissolved in 96% ethanol. Biofilm growth was quantified by measuring the OD at 560 nm using a microplate reader (Infinite^®^ 200 PRO, Tecan, Männedorf, Switzerland). A positive control was bacteria without compounds. All experiments were carried out in three repetitions.

#### 3.1.4. Blood Bacteriostatic Activity Tests

The assay was performed as previously described by [49] with a modification. Pure sheep blood was used for the research. Two strains of bacteria *S. aureus* ATCC 6538 and *S. epidermidis* ATCC 14990 were used with an inoculum density of approximately 108 CFU/mL at 0 time. The final concentration of the compounds was 0.5, 1, 2, 4, and 8 µg/mL, and the final volume of the assay tube was 1 mL. Bacterial survivors were determined on MH agar plates (CFU/mL) at 24 h after exposure.

### 3.2. In Vitro Cytotoxicity Studies

#### 3.2.1. Hemolysis Assay

Sterile, defibrinated sheep blood was diluted in the ratio of 1:9 *v*/*v* with PBS (pH was 7) and 0.5 mL of diluted blood was pipetted in different 1.5 mL Eppendorf tubes. Compounds at a concentration of 125 to 0.5 µg/mL were added to the diluted blood samples, and these samples were incubated at 37 °C. Triton–x 100 (1%, *v*/*v*) treated with a blood sample was considered as a positive control. After incubating for 24 h, the tubes were centrifuged at 3000 rpm for 10 min at room temperature. The amount of 200 μL of the supernatants was collected into a 96-well microtiter plate, and the absorbance was measured at 540 nm using a microtiter plate reader (Infinite^®^ 200 PRO, Tecan) [50].

#### 3.2.2. Cell Culture

The human cervical cancer HeLa, ovarian cancer SKOV-3, breast cancer MCF-7, and noncancerous embryonic kidney HEK-293 cells were obtained from the American Type Culture Collection (ATCC, Manassas, VA, USA). The HeLa, MCF-7, and HEK-293 cells were cultured in Dulbecco’s modified Eagle’s medium (DMEM). The SKOV-3 cells were maintained in McCoy’s medium. Both media were supplemented with 100 units/mL of penicillin, 100 µg/mL of streptomycin, and 10% (*v*/*v*) fetal bovine serum (FBS) (Merck Millipore, Burlington, MA, USA). The cells were incubated at 37 °C and 5% CO_2_.

#### 3.2.3. MTT Assay

MTT [3–(4,5–dimethylthiazol–2–yl)–2,5–diphenyltetrazolium bromide] assay was used to estimate the cytotoxic activity of 10 compounds. Vinblastine sulphate was used in this assay as a positive control. All the cell lines were seeded in 96-well plates at a density of 5 × 10^3^ cells/well and treated for 24 h with the compounds at concentrations of 2.5–150.0 µg/mL. After treatment, the cells were incubated with MTT (0.5 mg/mL; Merck Millipore, Burlington, MA, USA) for 3 h. The formed formazan crystals were dissolved in DMSO. To measure the absorbance of the formazan solution, a plate reader (Epoch, BioTek Instruments, Santa Clara, CA, USA) was used. The results [±standard deviation (SD)] were obtained from six repetitions in at least two independent experiments. The data are expressed as IC_50_ values (µg/mL).

### 3.3. Antioxidant

#### 3.3.1. Materials

DPPH (2,2–diphenyl–1–picrylhydrazyl), ABTS [2,2′–azino–*bis*(3–ethylbenzthiazoline–6–sulfonic acid)], potassium persulfate DMSO (dimethyl sulfoxide), and ascorbic acid were sourced from Sigma Chemical Co. (St. Louis, MO, USA), HPLC-grade methanol was purchased from P.O.Ch. (Gliwice, Poland).

#### 3.3.2. DPPH Assay

The DPPH assay was performed spectrophotometrically [51,52]. The amount of 30 µL of different concentrations of samples, dissolved in methanol/DMSO (9/1 *v*/*v*), was mixed with 170 µL of 0.06 mM DPPH methanolic solution and completed with methanol to a final volume of 300 µL. DPPH methanolic solution with methanol/DMSO (9/1 *v*/*v*) was used as a control. After 40 min of incubation at room temperature in the dark, the absorbance was analyzed at λ = 510 nm by a 96-well microplate reader (Epoch, BioTek System, Santa Clara, CA, USA). Ascorbic acid was used as a standard.

DPPH inhibition was calculated by the following equation:DPPH Inhibition (%) = [Acontrol − Asample)/Acontrol] × 100%

IC_50_ values of the samples were calculated using program GraFit v. 7.0 (Erithacus Software). The analysis was performed three times, with tree replications in each (*n* = 9). Statistical analysis was performed with one-way ANOVA with Tuckey’s post-hoc test.

#### 3.3.3. ABTS Assay

The ABTS assay was performed spectrophotometrically [51,52]. The amount of 30 µL of different concentrations of samples, dissolved in methanol/DMSO (9/1 *v*/*v*), was mixed with 170 µL of ABTS solution (2 mM ABTS diammonium salt, 3.5 mM potassium persulfate) and completed with water to a final volume of 300 µL. ABTS solution with methanol/DMSO (9/1 *v*/*v*) was used as a control. After 10 min of incubation at 30 °C in the dark, the absorbance was analyzed at λ = 750 nm by a 96-well microplate reader (Epoch, BioTek System, Santa Clara, CA, USA). Ascorbic acid was used as a standard.

ABTS inhibition was calculated by the following equation:ABTS Inhibition (%) = [Acontrol − Asample)/Acontrol] × 100%

IC_50_ values of the samples were calculated using program GraFit v. 7.0 (Erithacus Software). The analysis was performed three times, with tree replications in each (*n* = 9). Statistical analysis was performed with one-way ANOVA with Tuckey’s post-hoc test.

### 3.4. Synthesis

The following instruments and parameters were used: melting points Boetius PHMK apparatus; IR spectra: KBr pellets, 400–4000 cm^–1^ Thermo Mattson Satellite FTIR spectrophotometer; ^1^H and ^13^C NMR: Varian Unity 500 plus apparatus at 500 MHz; chemical shifts are expressed in parts per million (ppm) relative to TMS as internal standard. Elemental analyses for C, H, and N were performed on a 2400 Series II CHN Elemental Analyzer (Perkin Elmer, Shelton, CT, USA) and are in agreement with the theoretical values within ±0.4% range. Thin-layer chromatography (TLC) was performed on Merck Kieselgel 60 F_254_ plates and visualized with UV. Gravity column chromatography was performed on Fluka silica gel 60 of particle size 35–75 μm and 220–440 mesh (Sigma–Aldrich Chemie, Steinheim, Germany). The starting compounds (4–chloro–2–mercapto–5-methylbenzenesulfonamide (**1**)) [33] and cinnamic acid derivatives (**10**–**15**) were synthesized according to the literature sources [34,35,36].

#### 3.4.1. 4–Chloro–5–methyl–2–[(naphthalen–1–yl)methylthio]benzenesulfonamide (**2**)

A solution of 4–chloro–2-mercapto–5–methylbenzenesulfonamide 0.238 g (1 mmol) and 10 mL of CH_2_Cl_2_, 0.15 mL of triethylamine (1.07 mmol) was placed in an ice-water bath and stirred for 5 min. Then, 194 mg (1.1 mmol) of 1–(chloromethyl)naphthalene chloride was added and stirred at room temperature for 24 h. The solid precipitated was filtered and crystallized from ethanol to produce pure product **2** (0.215 g, 57%) as a white solid: mp 177–179 °C; IR (KBr) ν_max_ 3356, 3260, 3061, 2955, 2922, 2855, 1587, 1531, 1348, 1158 cm^–1^; ^1^H NMR (DMSO-*d*_6_, 500 MHz) δ: 2.34 (s, 3H, CH_3_), 4.81 (s, 2H, CH_2_S), 7.33 (s, 2H, NH_2_), 7.44–7.47 (m, 1H, Ar), 7.56–7.62 (m, 3H, Ar), 7.63 (s, 1H, H-3), 7.83 (s, 1H, H-6), 7.87–7.91 (m, 1H, Ar), 7.96 (d, 1H, Ar, *J* = 7.8 Hz), 8.25 (d, 1H, Ar, *J* = 8.3 Hz) ppm; Anal. Calcd for C_18_H_16_ClNO_2_S_2_ (377.91): C, 57.21; H, 4.27; N, 3.71; found: C, 57.20; H, 4.25; N, 3.69.

#### 3.4.2. 4–Chloro–2-[(6–chlorobenzo[*d*][1,3]dioxol–5–yl)methylthio]–5–methylbenzenesulfonamide (**3**)

A solution of 4–chloro–2–mercapto–5–methylbenzenesulfonamide 0.238 g (1 mmol) and 10 mL of CH_2_Cl_2_, 0.15 mL of triethylamine (1.07 mmol) was placed in an ice–water bath and stirred for 5 min. Then, 0.225 g (1.1 mmol) of 6–chloropiperonyl chloride was added and stirred at room temperature for 24 h. The precipitated solid was filtered and crystallized from ethanol to produce pure product **3** (0.329 g, 81%) as a white solid: mp 179–180 °C; IR (KBr) ν_max_ 3331, 3241, 3108, 3051, 2905, 1478, 1470, 1351, 1160 cm^–1^; ^1^H NMR (DMSO-*d*_6_, 500 MHz) δ: 2.33 (s, 3H, CH_3_), 4.30 (s, 2H, CH_2_S), 6.05 (s, 2H, O–CH_2_–O), 7.10 (s, 1H, Ar), 7.12 (s, 1H, Ar), 7.37 (s, 2H, NH_2_), 7.49 (s, 1H, H-3), 7.83 (s, 1H, H-6) ppm; Anal. Calcd for C_15_H_13_Cl_2_NO_4_S_2_ (406.30): C, 44.34; H, 3.22; N, 3.45; found: C, 44.32; H, 3.22; N, 3.39.

#### 3.4.3. Synthesis of N–{[4–chloro–5–methyl–2–(naphthalen–1–ylmethylthio)phenyl]sulfonyl}cinnamamide derivatives (**16a–16f**)

A mixture of 0.75 mmol (0.283 g) of 4–chloro–5–methyl–2–[(naphthalen–1–yl)methylthio]benzenesulfonamide (**2**), 0.9 mmol of EDCI (0.172 g), and 1.55 mmol of DMAP (0.189 g) dissolved in 11 mL of dichloromethane followed by dropwise addition of a solution of the appropriate cinnamic acid derivative (0.9 mmol in 4 mL of CH_2_Cl_2_). The whole was stirred at room temperature for 24 h. Then, 10 mL of water was added to the solution and acidified with 2M HCl solution to pH~2. The precipitate was filtered.

##### *N*–[4–Chloro–5–methyl–2–(naphthalen–1–ylmethylthio)benzenesulfonyl]cinnamamide (**16a**) [29,30]

This product was purified by column chromatography with methylene chloride as eluent, followed by crystallization from ethanol. The compound was obtained as a colorless solid in 51% yield (0.195 g): mp 105–107 °C; IR (KBr) ν_max_ 3196, 3062, 2921, 2858, 1686, 1530, 1450, 1352, 1178 cm^–1^; ^1^H NMR (DMSO-*d*_6_, 500 MHz) δ: 2.49 (s, 3H, CH_3_), 4.87 (s, 2H, CH_2_S), 6.56 (d, 1H, C**H**=CH–CO, *J* = 16.1 Hz), 7.35–7.41 (m, 3H, Ar), 7.47–7.48 (m, 3H, Ar), 7.53–7.56 (m, 4H, Ar and CH=C**H**–CO), 7.75 (s, 1H, H-3), 7.84 (d, 1H, Ar, *J* = 8.3 Hz), 7.89 (d, 1H, Ar, *J* = 7.8 Hz), 8.00 (s, 1H, H-6), 8.16 (d, 1H, Ar, *J* = 8.3 Hz), 12.38 (s, 1H, NHSO_2_) ppm; Anal. Calcd for C_27_H_22_ClNO_3_S_2_ (508.05): C, 63.83; H, 4.36; N, 2.76; found: C, 63.83; H, 4.32;N, 2.74.

##### (*E*)–3–(4–Bromophenyl)–*N*–{[4–chloro–5–methyl–2–(naphthalen–1–yl)methylthio]phenylsulfonyl}acrylamide (**16b**)

This product was purified by column chromatography with methylene chloride as eluent, followed by crystallization from ethanol. The compound was obtained as a colorless solid in 39% yield (0.170 g): mp 123–125 °C; IR (KBr) ν_max_ 3228, 3064, 2921, 2858, 1699, 1488, 1446, 1351, 1177 cm^–1^; ^1^H NMR (DMSO-*d*_6_, 500 MHz) δ: 2.49 (s, 3H, CH_3_), 4.87 (s, 2H, CH_2_S), 6.55 (d, 1H, C**H**=CH–CO, *J* = 15.6 Hz), 7.35–7.42 (m, 3H, Ar), 7.48 (d, 2H, Ar, *J* = 8.3 Hz), 7.51 (d, 1H, CH=C**H**–CO, *J* = 15.6 Hz), 7.55 (d, 1H, Ar, *J* = 6.8 Hz), 7.68 (d, 2H, Ar, *J* = 8.3 Hz), 7.75 (s, 1H, H-3), 7.85 (d, 1H, Ar, *J* = 8.3 Hz), 7.89 (d, 1H, Ar, *J* = 7.8 Hz), 7.99 (s, 1H, H-6), 8.15 (d, 1H, Ar, *J* = 8.3 Hz), 12.41 (s, 1H, NHSO_2_) ppm; Anal. Calcd for C_27_H_21_BrClNO_3_S_2_ (586.95): C, 55.25; H, 3.61; N, 2.39; found: C, 55.30; H, 3.62; N, 2.32.

##### (*E*)–*N*–{4–Chloro–5–methyl–2–[(naphthalen–1–yl)methylthio]phenylsulfonyl}–3–(4–fluorophenyl)acrylamide (**16c**)

This product was purified by column chromatography with methylene chloride–methanol (50:1) as eluent, followed by crystallization from 2-propanol. The compound was obtained as a colorless solid in 56% yield (0.248 g): mp 205–207 °C; IR (KBr) ν_max_ 3199, 3049, 2967, 2856, 1699, 1624, 1510, 1449, 1352, 1178 cm^–1^; ^1^H NMR (DMSO-*d*_6_, 500 MHz) δ: 2.39 (s, 3H, CH_3_), 4.87 (s, 2H, CH_2_S), 6.48 (d, 1H, C**H**=CH–CO, *J* = 16.1 Hz), 7.30–7.42 (m, 5H, Ar), 7.54 (d, 1H, CH=C**H**=CO, *J* = 15.6 Hz), 7.55–7.56 (m, 1H, Ar), 7.58–7.61 (m, 2H, Ar), 7.75 (s, 1H, H-3), 7.84 (d, 1H, Ar, *J* = 7.8 Hz), 7.90 (d, 1H, Ar, *J* = 9.8 Hz), 8.00 (s, 1H, H-6), 8.15 (d, 1H, Ar, *J* = 8.3 Hz), 12.37 (s, 1H, NHSO_2_) ppm; Anal. Calcd for C_27_H_21_ClFNO_3_S_2_ (526.04): C, 61.65; H, 4.02; N, 2.66; found: C, 61.63; H, 4.00; N, 2.62.

##### (*E*)–*N*–{4–Chloro–5–methyl–2–[(naphthalen–1–yl)methylthio]phenylsulfonyl}–3–(4–chlorophenyl)acrylamide (**16d**)

This product was purified by column chromatography with methylene chloride as eluent, followed by crystallization from ethanol. The compound was obtained as a colorless solid in 59% yield (0.238 g): mp 218–220 °C; IR (KBr) ν_max_ 3207, 3062, 2966, 2858, 1702, 1625, 1510, 1449, 1353, 1178 cm^–1^; ^1^H NMR (DMSO-*d*_6_, 500 MHz) δ: 2.49 (s, 3H, CH_3_), 4.88 (s, 2H, CH_2_S), 6.53 (d, 1H, C**H**=CH–CO, *J* = 15.7 Hz), 7.35–7.42 (m, 3H, Ar), 7.52 (d, 1H, CH=C**H**–CO, *J* = 16.6 Hz), 7.53–7.57 (m, 5H, Ar), 7.75 (s, 1H, H-3), 7.84 (d, 1H, Ar, *J* = 8.3 Hz), 7.89 (d, 1H, Ar, *J* = 7.8 Hz), 8.00 (s, 1H, H-6), 8.15 (d, 1H, Ar, *J* = 8.3 Hz), 12.40 (s, 1H, NHSO_2_) ppm; Anal. Calcd for C_27_H_21_Cl_2_NO_3_S_2_ (542.50): C, 59.78; H, 3.90; N, 2.58; found: C, 59.76; H, 3.89; N, 2.52.

##### (*E*)–*N*–{4–Chloro–5–methyl–2–[(naphthalen–1–yl)methylthio]phenylsulfonyl}–3–(3–fluoro–4–methoxyphenyl)acrylamide (**16e**)

This product was purified by column chromatography with methylene chloride as eluent, followed by crystallization from 2-propanol. The compound was obtained as a colorless solid in 40% yield (0.186 g): mp 153–155 °C; IR (KBr) ν_max_ 3152, 3102, 2972, 2934, 1665, 1603, 1514, 1455, 1351, 1182 cm^–1^; ^1^H NMR (DMSO-*d*_6_, 500 MHz) δ: 2.49 (s, 3H, CH_3_), 3.90 (s, 3H, OCH_3_), 4.87 (s, 2H, CH_2_S), 6.41 (d, 1H, C**H**=CH–CO, *J* = 15.6 Hz), 7.26 (t, 1H, Ar, *J_1,2_* = 8.6 Hz), 7.35–7.44 (m, 5H, Ar), 7.46 (d, 1H, CH=C**H**–CO, *J* = 16.1 Hz), 7.56 (d, 1H, Ar, *J* = 6.9 Hz), 7.73 (s, 1H, H-3), 7.85 (d, 1H, Ar, *J* = 7.8 Hz), 7.90 (d, 1H, Ar, *J* = 7.8 Hz), 7.99 (s, 1H, H-6), 8.15 (d, 1H, Ar, *J* = 8.3 Hz), 12.30 (s, 1H, NHSO_2_) ppm; ^13^C NMR (DMSO-*d*_6_, 125 MHz) δ 19.46, 34.95, 56.65, 114.52, 115.26, 115.40, 117.93, 124.61, 125.87, 126.27, 127.43, 128.56, 128.86, 131.52, 131.79, 132.84, 133.87, 134.07, 134.63, 137.13, 139.53, 143.10, 149.70, 150.89, 152.85, 163.65 ppm; Anal. Calcd for C_28_H_23_ClFNO_4_S_2_ (556.07): C, 60.48; H, 4.17; N, 2.52; found: C, 60.47; H, 4.15; N, 2.49.

##### (*E*)–*N*–{4–Chloro–5–methyl–2–[(naphthalen–1–yl)methylthio]phenylsulfonyl}–3–(4–nitrophenyl)acrylamide (**16f**)

This product was purified by column chromatography with methylene chloride–methanol (50:1) as eluent, followed by crystallization from ethanol. The compound was obtained as a yellow solid in 56% yield (0.230 g): mp 210–212 °C; IR (KBr) ν_max_ 3221, 3110, 2916, 2879, 1707, 1632, 1517, 1436, 1344, 1177 cm^–1^; ^1^H NMR (DMSO-*d*_6_, 500 MHz) δ: 2.49 (s, 3H, CH_3_), 4.88 (s, 2H, CH_2_S), 6.67 (d, 1H, C**H**=CH–CO, *J* = 16.1 Hz), 7.35–7.40 (m, 3H, Ar), 7.55 (d, 1H, CH=C**H**–CO, *J* = 15.6 Hz), 7.63 (d, 1H, Ar, *J* = 7.4 Hz), 7.77 (s, 1H, H-3), 7.80–7.88 (m, 4H, Ar), 8.00 (s, 1H, H-6), 8.15 (m, 1H, Ar), 8.30 (d, 2H, Ar, *J* = 8.8 Hz), 12.54 (s, 1H, NHSO_2_) ppm; ^13^C NMR (DMSO-*d*_6_, 125 MHz) δ 19.46, 34.93, 123.14, 124.57, 124.67, 125.86, 126.36, 126.76, 128.53, 128.62, 128.87, 128.92, 129.68, 131.52, 131.73, 132.94, 133.86, 134.10, 134.40, 137.15, 139.70, 140.60, 141.74, 148.65, 163.08 ppm; Anal. Calcd for C_27_H_21_ClN_2_O_5_S_2_ (553.05): C, 58.64; H, 3.83; N, 5.07; found: C, 58.57; H, 3.80; N, 4.98.

#### 3.4.4. Synthesis of *N–*{[4–chloro–2–(6–chlorobenzo[*d*][1,3]dioxol–5–yl)methylthio-5–methylphenyl]sulfonyl}cinnamamide derivatives (**17a**–**17d**)

A mixture of 0.75 mmol (0.305 g) of 4–chloro–2–[(6–chlorobenzo[*d*][1,3]dioxol–5–yl)methylthio]-5-methylbenzenesulfonamide (**3**), 0.9 mmol of EDCI (0.172 g), and 1.55 mmol of DMAP (0.189 g) was dissolved in 11 mL of dichloromethane followed by dropwise addition of a solution of the appropriate cinnamic acid derivative (0.9 mmol in 4 mL of CH_2_Cl_2_). The whole was stirred at room temperature for 24 h. Then, 10 mL of water was added to the solution and acidified with 2M HCl solution to pH~2. The precipitate was filtered.

##### (*E*)-3-(4-Bromophenyl)-*N*-{4-chloro-2-[(6-chlorobenzo[*d*][1,3]dioxol-5-yl)methylthio]-5-methylphenylsulfonyl}acrylamide (**17a**)

This product was purified by column chromatography with methylene chloride as eluent, followed by crystallization from ethanol. The compound was obtained as a colorless solid in 39% yield (0.178 g): mp 193–195 °C; IR (KBr) ν_max_ 3321, 3101, 2911, 1707, 1530, 1437, 1349, 1169 cm^–1^; ^1^H NMR (DMSO-*d*_6_, 500 MHz) δ: 2.38 (s, 3H, CH_3_), 4.35 (s, 2H, CH_2_S), 5.95 (s, 2H, O–CH_2_–O), 6.66 (d, 1H, C**H**=CH–CO, *J* = 15.6 Hz), 6.98 (s, 1H, Ar), 7.03 (s, 1H, Ar), 7.49 (d, 1H, CH=C**H**–CO, *J* = 16.1 Hz), 7.50 (d, 2H, Ar, *J* = 8.2 Hz), 7.61 (s, 1H, H-3), 7.65 (d, 2H, Ar, *J* = 9.1 Hz), 7.99 (s, 1H, H-6), 12.50 (s, 1H, NHSO_2_) ppm; ^13^C NMR (DMSO-*d*_6_, 125 MHz) δ: 19.46, 35.19, 102.53, 110.00, 110.79, 119.84, 124.47, 125.87, 126.61, 128.88, 130.49, 132.52, 133.29, 133.58, 134.21, 135.14, 136.17, 139.55, 143.05, 147.05, 148.15, 163.52 ppm; Anal. Calcd for C_24_H_18_BrCl_2_NO_5_S_2_ (615.34): C, 46.84; H, 2.95; N, 2.28; found: C, 46.83; H, 2.92; N, 2.26.

##### (*E*)–*N*–{4–Chloro–2–[(6–chlorobenzo[*d*][1,3]dioxol–5–yl)methylthio]–5–methylphenylsulfonyl}–3–(4–fluorophenyl)acrylamide (**17b**)

This product was purified by column chromatography with methylene chloride as eluent, followed by crystallization from 2-propanol. The compound was obtained as a colorless solid in 63% yield (0.261 g): mp 180–182 °C; IR (KBr) ν_max_ 3280, 3099, 2972, 2901, 1705, 1478, 1451, 1351, 1174 cm^–1^; ^1^H NMR (DMSO-*d*_6_, 500 MHz) δ: 2.38 (s, 3H, CH_3_), 4.35 (s, 2H, CH_2_S), 5.94 (s, 2H, O–CH_2_–O), 6.59 (d, 1H, C**H**=CH–CO, *J* = 15.6 Hz), 6.98 (s, 1H, Ar), 7.03 (s, 1H, Ar), 7.29 (t, 2H, Ar, *J_1,2_* = 8.8 Hz), 7.52 (d, 1H, CH=C**H**–CO, *J* = 16.1 Hz), 7.60–7.63 (m, 2H, Ar and s, 1H, H-3), 7.99 (s, 1H, H-6), 12.45 (s, 1H, NHSO_2_) ppm; ^13^C NMR (DMSO-*d*_6_, 125 MHz) δ: 19.46, 35.17, 102.52, 109.99, 110.81, 116.50, 116.68, 118.87, 125.86, 126.63, 128.89, 130.98, 133.28, 134.21, 135.18, 136.15, 139.52, 143.15, 147.06, 148.15, 162.82, 163.62, 164.80 ppm; Anal. Calcd for C_24_H_18_Cl_2_FNO_5_S_2_ (554.44): C, 51.99; H, 3.27; N, 2.53; found: C, 51.98; H, 3.25; N, 2.50.

##### (*E*)–*N*–{4–Chloro–2–[(6–chlorobenzo[*d*][1,3]dioxol–5–yl)methylthio]–5–methylphenylsulfonyl}–3–(4–chlorophenyl)acrylamide (**17c**)

This product was purified by column chromatography with methylene chloride as eluent, followed by crystallization from 2-propanol. The compound was obtained as a colorless solid in 58% yield (0.249 g): mp 185–187 °C; IR (KBr) ν_max_ 3368, 3102, 2915, 2852, 1706, 1635, 1529, 1437, 1350, 1170 cm^–1^; ^1^H NMR (DMSO-*d*_6_, 500 MHz) δ: 2.49 (s, 3H, CH_3_), 4.35 (s, 2H, CH_2_S), 5.94 (s, 2H, O–CH_2_–O), 6.64 (d, 1H, C**H**=CH–CO, *J* = 15.6 Hz), 6.98 (s, 1H, Ar), 7.03 (s, 1H, Ar), 7.50 (d, 1H, CH=C**H**–CO, *J* = 15.6 Hz), 7.52 (d, 2H, Ar, *J* = 8.3 Hz), 7.57 (d, 2H, Ar, *J* = 8.8 Hz), 7.61 (s, 1H, H-3), 7.99 (s, 1H, H-6), 12.50 (s, 1H, NHSO_2_) ppm; ^13^C NMR (DMSO-*d*_6_, 125 MHz) δ: 19.46, 35.20, 102.53, 110.00, 110.79, 119.72, 125.87, 126.61, 128.91, 129.59, 130.29, 133.23, 133.30, 134.23, 135.11, 135.61, 136.17, 139.58, 142.99, 147.05, 148.15, 163.49 ppm; Anal. Calcd for C_24_H_18_Cl_3_NO_5_S_2_ (570.89): C, 50.49; H, 3.18; N, 2.45; found: C, 50.49; H, 3.15; N, 2.40.

##### (*E*)–*N*–{4–Chloro–2–[(6–chlorobenzo[*d*][1,3]dioxol–5–yl)methylthio]–5–methylphenylsulfonyl}–3–(4–nitrophenyl)acrylamide (**17d**)

This product was purified by column chromatography with methylene chloride as eluent, followed by crystallization from 2-propanol. The compound was obtained as a colorless solid in 64% yield (0.280 g): mp 106–108 °C; IR (KBr) ν_max_ 3268, 3109, 2981, 2902, 1719, 1633, 1519, 1476, 1344, 1172 cm^–1^; ^1^H NMR (DMSO-*d*_6_, 500 MHz) δ: 2.38 (s, 3H, CH_3_), 4.36 (s, 2H, CH_2_S), 5.92 (s, 2H, O–CH_2_–O), 6.80 (d, 1H, C**H**=CH–CO, *J* = 16.1 Hz), 6.97 (s, 1H, Ar), 7.02 (s, 1H, Ar), 7.62 (s, 1H, H-3), 7.63 (d, 1H, CH=C**H**–CO, *J* = 15.6 Hz), 7.81 (d, 2H, Ar, *J* = 8.8 Hz), 8.00 (s, 1H, H-6), 8.28 (d, 2H, Ar, *J* = 8.3 Hz), 12.63 (s, 1H, NHSO_2_) ppm; Anal. Calcd for C_24_H_18_Cl_2_N_2_O_7_S_2_ (581.44): C, 49.58; H, 3.12; N, 4.82; found: C, 49.56; H, 3.10; N, 4.79.

### 3.5. Evaluation of Maximum Absorbance of the Compounds

The maximum absorbance of ten compounds was evaluated with a multi-plate reader Epoch (Epoch, BioTek Instruments, Santa Clara, CA, USA). The compounds were dissolved in 10% DMSO in methanol (*v*/*v*) or were mixed with cell culture medium, and the absorption was measured at time t_0_ and after 96 h. The values of the maximum absorbance of the compounds are presented in Appendix A.

## 4. Conclusions

We have developed a method for the synthesis of a new series of *N*–(4–chloro–2–mercapto–5-methylphenylsulfonyl)cinnamamide derivatives containing two pharmacophore groups: cinnamic acid and benzenesulfonamide. We have evaluated their antimicrobial, anticancer, and antioxidant activity. All performed experiments showed that chemical modifications allowed us to obtain derivatives with high activity against the tested clinical strains. All of the compounds were active on *Staphylococcus* and *Enterococcus* species. However, compounds **16b**, **16d**, **16e,** and **16f** containing a naphthylmethylthiol substituent were the most active on *Staphylococcus* MRSA and MRCNS strains. On the other hand, in experiments with clinical strains of *Enterococcus* sp., the compounds showed varying activity, with compounds **16a** and **16c** being the most active. Among these derivatives, compounds **16d** and **16e** had the best inhibition properties against biofilm formation and biofilm already formed by MRCNS strains. The hemolysis test revealed that concentrations of the compounds greater than 32 µg/mL caused an interaction with peripheral blood erythrocytes, leading to hemolysis. Compounds **16c**, **16d**, **17a,** and **17d** of the tested group showed very moderate in vitro activity against human tumor cell lines. In antiradical tests, compounds **16f** and **17d** demonstrated the strongest antioxidant properties. Cinnamic acid derivatives **16d** and **17d** revealed both anticancer properties and minor antiradical activity. On the other hand, compound **16f** was observed to show promising antioxidant activity, while little activity against tumor lines was observed.

The structure–activity relationship analysis indicates that in the case of *N*–(4–chloro–2–mercapto–5–methylphenylsulfonyl)cinnamamide derivatives, the presence of 1–naphthyl in R^2^ position determines their significant activity against bacteria strains, which can help guide the future design of similar antibacterial compounds. In the experiments with cancer cells, the results show that the presence of fluorine and chlorine in R^1^ and 1–naphthyl in R^2^, as well as bromine and nitro group in R^1^ and 6–chloropiperonyl in R^2^, can be responsible for stronger anticancer activity of cinnamamide derivatives. On the other hand, the presence of nitro group in R^1^ position and the acylvinyl group seems to be essential for the antioxidant activity of the hybrids.

In summary, our results indicate that among the tested activities of the cinnamic acid derivatives, the antibacterial effect is the most promising in the medical use and treatment of bacterial infections.

## Data Availability

Not applicable.

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
