# Peer review of "An In Vitro Antimicrobial, Anticancer and Antioxidant Activity of N–[(2–Arylmethylthio)phenylsulfonyl]cinnamamide Derivatives"

_molecules, 2023, doi:10.3390/molecules28073087_

Round 1

Reviewer 1 Report

BuÅ‚akowska et al. report the synthesis of new series of N-(4-chloro-2-mercapto-5-methylphenylsulfonyl)cinnamamide derivatives containing two pharmacophore groups: cinnamic acid and benzenesulfonamide. Then they evaluated  their antimicrobial, anticancer and antioxidant activities. It is obvious, that this work is sound and  of interest for the readers of "Molecules". Nevertheless, there is some points to be fixed before the publication of this paper.  

1- The quality of figures should be improved.

2- Althought several biological activities were evaluated, the structure activity relationship was not disscused. 

3- In the "Materials and methods" section, it is preferable to begin with the synthesis part.

Author Response

Dear Editor,

We would like to thank for critical reading this manuscript and valuable suggestions. We have carefully considered all of the suggestions and made the appropriate additions. The corrections are red in color.

In the response to Reviewer #1:

Comment 1: The quality of figures should be improved.

Response: The quality of figures has been improved.

Comment 2: Althought several biological activities were evaluated, the structure activity relationship was not disscused.

Response: Thank you for this comment. The structure-activity relationship discussion has been added and it is in lines – 295-308, 439-445, 605-610, and also in Conclusion section (lines 1020-1028).   

Comment 3: In the "Materials and methods" section, it is preferable to begin with the synthesis part.

Response: The synthesis part in the "Materials and methods" section was placed at the end, because we followed the similar articles in the Special Issue “Antioxidant, Antimicrobical and Anticancer Activity of Natural Product and Their Derivatives” with the synthesis part, thus we have planned our paper the same.

We thank the Reviewers for all the suggestions and hope that the revised manuscript is now appropriate for publication in Molecules, Section Natural Products Chemistry, Special Issue “Antioxidant, Antimicrobical and Anticancer Activity of Natural Product and Their Derivatives”.

Sincerely,

Anita Bułakowska, Ph.D.

Assistant Professor

Department of Organic Chemistry

Medical University of Gdańsk

Al. Gen. J. Hallera 107, 80-416 Gdańsk, Poland

Reviewer 2 Report

The manuscript entitled "An in vitro antimicrobial, anticancer and antioxidant activity of N-[(2-arylmethylthio)phenylsulfonyl]cinnamamide derivatives" is a good work. However, I recommend a major revision prior to further consideration.

1. Abstract should contain units for all the mentioned values.

2. The IC50 values are found to be at very high doses, I hope there require an explanation for this reduced activity. I suggest to describe and support this in terms of a nutraceutical, which can be obtained from food sources at large quantities. Otherwise, as a pharmacological drug, it should show high activity.

3. In the introduction I suggest to explain the details of pharmacological effects of cinnamic acid or its derivatives as well as its sources (with quantities)

4. I also suggest to include outline on the advancements in cinnamic acid in terms of clinical reports as well as patents

5. In the antimicrobial activity, authors could have estimated the zone of inhibition by disc diffusion method

6. Combining Result and Discussion section made it difficult for in depth analysis of the available results.  Hence, I suggest to split it as two separate sections.

7. I suggest to include the details of supplementary materials in manuscript

Author Response

Dear Editor,

We would like to thank for critical reading this manuscript and valuable suggestions. We have carefully considered all of the suggestions and made the appropriate additions. The corrections are red in color.

In the response to Reviewer #2:

Comment 1: Abstract should contain units for all the mentioned values.

Response: The units for mentioned values have been added.

Comment 2: The IC50 values are found to be at very high doses, I hope there require an explanation for this reduced activity. I suggest to describe and support this in terms of a nutraceutical, which can be obtained from food sources at large quantities. Otherwise, as a pharmacological drug, it should show high activity.

Response: Ten compounds were evaluated with antibacterial, anticancer and antioxidant assays. Among the tested activities of the cinnamamide derivatives, the antibacterial effect is the most promising in the medical use. We are going on focus on this activity, not on anticancer and antioxidant, where these compounds did not show promising results. We agree that their IC50 values were not significant. Although the tested derivatives have cinnamic acid - naturally occurring plant metabolite, the hybrids with sulfonamides are synthetic, so it can not be obtained from food sources.

We will continue the study with the selected compounds and their practical application rather on skin as the antimicrobial medicaments or disinfectants due to their great activity on Staphylococcus aureus and S. epidermidis

Comment 3: In the introduction I suggest to explain the details of pharmacological effects of cinnamic acid or its derivatives as well as its sources (with quantities).

Response: According to the Reviewer’s suggestion the Introduction section has been rewritten (lines 83-91).

Comment 4: I also suggest to include outline on the advancements in cinnamic acid in terms of clinical reports as well as patents.

Response: Clinical reports of cinnamic acid derivatives have been added to the text (lines 103-112).

Comment 5: In the antimicrobial activity, authors could have estimated the zone of inhibition by disc diffusion method.

Response: The tested compounds show hydrophobic properties and diffuse poorly in the agar medium, giving small zones of growth inhibition. We decided that the results obtained by the disc diffusion method do not bring anything significant to our research and we decided not to show them.

Comment 6: Combining Result and Discussion section made it difficult for in depth analysis of the available results. Hence, I suggest to split it as two separate sections.

Response: We have combined the Result and Discussion section due to the different results and activities of the compounds that we have obtained. For better understanding as well as discussion of the results in each section we decided to combine these two sections. In other case, in our opinion, it would be difficult to present the discussion of such different effects and show this in a clear way. While the conclusion section has been expanded and the most important results have been collected here.     

Comment 7: I suggest to include the details of supplementary materials in manuscript.

Response: Due to the fact that the manuscript is very comprehensive, some research results have been included in the supplementary materials. However in the revised manuscript we have put the biological results in Tables in the main text, as suggested.  

We thank the Reviewers for all the suggestions and hope that the revised manuscript is now appropriate for publication in Molecules, Section Natural Products Chemistry, Special Issue “Antioxidant, Antimicrobical and Anticancer Activity of Natural Product and Their Derivatives”.

Sincerely,

Anita Bułakowska, Ph.D.

Assistant Professor

Department of Organic Chemistry

Medical University of Gdańsk

Al. Gen. J. Hallera 107, 80-416 Gdańsk, Poland

Reviewer 3 Report

The authors A. BuÅ‚akowska and co-workers submitted their manuscript entitled “An in vitro antimicrobial, anticancer and antioxidant activity of N-[(2-arylmethyl thio)phenylsulfonyl]cinnamamide derivatives” to the journal “MDPI” in order to be considered for publication as an “Article”.

The research work reports on the chemical synthesis and analytical characterization (IR, 1H/13C NMR, CHN analysis) of ten compounds bearing cinnamic acid and benzenesulfonamide cores serving as potential pharmacophores. The compounds were comprehensively investigated regarding antibacterial (also including resistant strains), antifungal, anti-biofilm, anticancer, and antioxidative properties.

I personally find the introduction a bit long and that it contains partly unnecessary information. I would put more focus on the need for the development of novel antibacterial drugs due to the multiple resistance situations. These arise, for example, through different mechanisms (mutations resulting in multidrug resistances 10.3390/molecules24173152, efflux of drug by pumps 10.1186/s12866-021-02250-x, loss of porin protein and reduced uptake of antibiotic drug 10.1128/spectrum.00148-22, among other mechanisms https://doi.org/10.1016/j.micpath.2021.104915). Therefore, it also makes sense to design compounds that contain different pharmacophores as hybrid drugs to circumvent different resistance pathways or cross-resistance. After all, this is actually the real big problem in connection with antibiotics research!

The figures are hard to read, especially the keys/labelling. Please revise.

How about the stability of the compounds in cell culture medium, for example for the cytotoxicity testing? The medium contains FBS, which itself contains esterases and amidases among others. Is the stability of the complexes warranted during the period of the biological experiments? Please provide experimental proof.

I am not so much into this aspect, but I kindly ask the authors to potentially consider this in their work. The general structure (Scheme 1) of their compounds comprises conjugated double bindings. The authors measure absorbance (e.g., MTT, DPPH, ABTS assay). Can absorption by the compounds be excluded? What is their absorption maximum?

How about the purity? Please provide values. Compounds undergoing biological experiments should generally have a purity >95%.

Can the authors please provide a kind of structure-activity-relationship in their discussion/conclusion. This could help guiding future design of similar compounds.

Formal aspects, e.g., subscript (e.g., Scheme 1, R1, 16f, NO2), superscript (e.g., caption Figure 7 and running text E+0X), line distance, different fonts among others.

All the best!

Author Response

Dear Editor,

We would like to thank for critical reading this manuscript and valuable suggestions. We have carefully considered all of the suggestions and made the appropriate additions.

In the response to Reviewer #3:

Comment 1: I personally find the introduction a bit long and that it contains partly unnecessary information. I would put more focus on the need for the development of novel antibacterial drugs due to the multiple resistance situations. These arise, for example, through different mechanisms (mutations resulting in multidrug resistances 10.3390/molecules24173152, efflux of drug by pumps 10.1186/s12866-021-02250-x, loss of porin protein and reduced uptake of antibiotic drug 10.1128/spectrum.00148-22, among other mechanisms https://doi.org/10.1016/j.micpath.2021.104915). Therefore, it also makes sense to design compounds that contain different pharmacophores as hybrid drugs to circumvent different resistance pathways or cross-resistance. After all, this is actually the real big problem in connection with antibiotics research!

Response: According to the Reviewer’s suggestion the Introduction section has been rewritten.

Comment 2: The figures are hard to read, especially the keys/labelling. Please revise.

Response: The quality of figures has been improved.

Comment 3: How about the stability of the compounds in cell culture medium, for example for the cytotoxicity testing? The medium contains FBS, which itself contains esterases and amidases among others. Is the stability of the complexes warranted during the period of the biological experiments? Please provide experimental proof.

Response: Thank you for this comment. In the time we had to respond to the reviews, we could only analyze differences in the λmax of the tested compounds in 10% DMSO/methanol (v/v) and in cell culture medium. We observed the stability of the tested compounds in cell culture medium for 96h. The observation was conducted for the first 6h in every 30min and afterwards in every 10h. The results at T0 and T96h are shown in Table S1 (Supplementary materials).

The absorbance analysis indicated that λmax of the compounds diluted in 10% DMSO/methanol (v/v) are in the range 304 – 324 nm, while in cell culture medium 297 – 305 nm. The hypsochromic shift in absorbance is not significant for compounds 16d, 16f and 17b (not higher than 10 nm) and can be caused by the change in viscosity in the probe. For the other compounds the shifts are in the range 15-20 nm and can be the result of both the change of viscosity and some quenching effect of macromolecules presented in cell culture medium. The observation of the stability of the compounds in cell culture medium were conducted for 96h and exhibited the stability of the tested compounds in the studied period of time. The maximum absorbance shifts between T0h and T96h for all the tested compounds are within the standard deviation.

The detailed studies on the stability of the presented compounds will be the subject of further in vivo research. Of course, the stability in vivo of the compounds may be completely different than in in vitro conditions.

We have also initially examined the selected compounds in program for prediction of pharmacokinetic properties of the compounds (https://biosig.lab.uq.edu.au/pkcsm/prediction) as the basis for our future study.

Comment 4: I am not so much into this aspect, but I kindly ask the authors to potentially consider this in their work. The general structure (Scheme 1) of their compounds comprises conjugated double bindings. The authors measure absorbance (e.g., MTT, DPPH, ABTS assay). Can absorption by the compounds be excluded? What is their absorption maximum?

Response: Antioxidant analyzes including DPPH and ABTS tests were carried out spectrophotometrically. The studies consisted of mixing a series of dilutions of the tested compounds with the appropriate DPPH or ABTS solution (in methanol and in water, respectively). The absorbance of the blank sample was subtracted from each obtained absorbance result. The blank sample contained the appropriate concentration of the tested compound dissolved in methanol or water (respectively for the DPPH and ABTS test).

In MTT assay, the absorbance of formazan crystals in DMSO is measured. The formazan forms in living cells from MTT solution, which is added to cells with medium. Before this step, compounds with medium are removed from the wells, and fresh medium without compounds is added to the wells with cells, according to  manufacturer’s protocol. Finally, only absorbance values of formazan dissolved in DMSO are obtained in this assay.

The absorption maxima of the compounds are shown in Table S1 (Supplementary materials).

Comment 5: How about the purity? Please provide values. Compounds undergoing biological experiments should generally have a purity >95%.

Response: The purity of the compounds was tested by measuring the melting point, elemental analysis and spectroscopic analysis (IR, NMR). In the 1H NMR spectra, only sharp signals confirming the structures of the compounds were observed. If number of sharp signals at the anticipated position matches with number of protons with correct splitting and integration, one can say the compound is pure.

Comment 6: Can the authors please provide a kind of structure-activity-relationship in their discussion/conclusion. This could help guiding future design of similar compounds.

Response: The structure-activity relationship discussion has been added (lines 295-308, 439-445, 605-610, and also in Conclusion section (lines 1020-1028).

Comment 7: Formal aspects, e.g., subscript (e.g., Scheme 1, R1, 16f, NO2), superscript (e.g., caption Figure 7 and running text E+0X), line distance, different fonts among others.

Response: As suggested by the Reviewer formal aspects have been changed.

We thank the Reviewers for all the suggestions and hope that the revised manuscript is now appropriate for publication in Molecules, Section Natural Products Chemistry, Special Issue “Antioxidant, Antimicrobical and Anticancer Activity of Natural Product and Their Derivatives”.

Sincerely,

Anita Bułakowska, Ph.D.

Assistant Professor

Department of Organic Chemistry

Medical University of Gdańsk

Al. Gen. J. Hallera 107, 80-416 Gdańsk, Poland

Round 2

Reviewer 2 Report

No more comments 

Author Response

Dear Editor,

we would like to thank for critical reading this manuscript and valuable suggestions. We have carefully considered all of the suggestions and made the appropriate additions. The changes are marked in yellow.

Reviewer 2:

Thank you for your review and your valuable suggestions. We improved as much as possible the manuscript and English.

We thank again the Reviewers for all the suggestions and hope that the revised manuscript is now appropriate for publication in Molecules, Section Natural Products Chemistry, Special Issue “Antioxidant, Antimicrobical and Anticancer Activity of Natural Product and Their Derivatives”.

Sincerely,

Anita Bułakowska, Ph.D.

Department of Organic Chemistry

Medical University of Gdańsk

Al. Gen. J. Hallera 107, 80-416 Gdańsk, Poland

Reviewer 3 Report

The authors Anita BuÅ‚akowska an co-workers provided a revised version of their manuscript “An in vitro antimicrobial, anticancer and antioxidant activity of N-[(2-arylmethylthio)phenylsulfonyl]cinnamamide derivatives” submitted to the journal “Molecules” in order to be considered for publication as an “Article”.

The authors have revised the introduction to focus on cinnamic acid and the urgent need to develop new antimicrobials. Such an orientation of the introduction now fits better to the embedding of the research work of the authors.

The authors performed additional experiments to study the stability of the compounds in the solvent respectively in cell culture medium. Stability could be confirmed. Regarding the purity, the authors rely on the spectroscopic characterization as well as melting points (well, there is no reference thus far). Additional experiments, e.g. using elemental (CHN) analysis or HPLC were not carried out. The authors base their conclusion about purity on the analytical characterization.

In my opinion, it is unusual and confusing to show the data presented in Figures 2, 3, and 4 in such a graph - especially by connecting the lines. This does not make sense. A table listing of the mean value (MIC, etc.) ± SD would make more sense, in my opinion.

Discussion of structure-activity-relationships were added to the manuscript, which may help guiding future design of similar drugs. Moreover, the authors improved the resolution of the figures. Formal errors have also been corrected; however, some are still present in the manuscript. Nevertheless, these inconsistencies will be ruled out during editing and proof-reading.

In conclusion, I suggest proceeding of the manuscript after this very minor revision. All the best!

Author Response

Dear Editor,

we would like to thank for critical reading this manuscript and valuable suggestions. We have carefully considered all of the suggestions and made the appropriate additions. The changes are marked in yellow.  

Reviewer 3:

Comment 1: The authors performed additional experiments to study the stability of the compounds in the solvent respectively in cell culture medium. Stability could be confirmed. Regarding the purity, the authors rely on the spectroscopic characterization as well as melting points (well, there is no reference thus far). Additional experiments, e.g. using elemental (CHN) analysis or HPLC were not carried out. The authors base their conclusion about purity on the analytical characterization.

Response 1: Elemental analysis (CHN) was performed for all the obtained compounds. It is described in the Results and Discussion chapter, point 2.1. Chemistry and in the Materials and Methods chapter, point 3.4. Synthesis, where each compound was analysed.

Comment 2: In my opinion, it is unusual and confusing to show the data presented in Figures 2, 3, and 4 in such a graph - especially by connecting the lines. This does not make sense. A table listing of the mean value (MIC, etc.) ± SD would make more sense, in my opinion.

Response 2: We have changed the type of graph in the case of Figures 2-4. We have showed the results from Figures 3 and 4 in one figure (now is Figure 3). We have also added the values of MIC (in Tables) to the Supplementary materials. 

Comment 3: Discussion of structure-activity-relationships were added to the manuscript, which may help guiding future design of similar drugs. Moreover, the authors improved the resolution of the figures. Formal errors have also been corrected; however, some are still present in the manuscript. Nevertheless, these inconsistencies will be ruled out during editing and proof-reading.

Response 3: We have corrected as much as possible the formal errors.

Sincerely,

Anita Bułakowska, Ph.D.

Department of Organic Chemistry

Medical University of Gdańsk

Al. Gen. J. Hallera 107, 80-416 Gdańsk, Poland